# In-Depth Longitudinal Comparison of Clinical Specimens to Detect SARS-CoV-2

**DOI:** 10.3390/pathogens10111362

**Published:** 2021-10-21

**Authors:** Justine Defêche, Samira Azarzar, Alyssia Mesdagh, Patricia Dellot, Amandine Tytgat, Fabrice Bureau, Laurent Gillet, Yasmine Belhadj, Sebastien Bontems, Marie-Pierre Hayette, Raphaël Schils, Souad Rahmouni, Marie Ernst, Michel Moutschen, Gilles Darcis

**Affiliations:** 1Department of Clinical Microbiology, University of Liège, 4000 Liège, Belgium; J.Defeche@chuliege.be (J.D.); sbontems@chuliege.be (S.B.); mphayette@chuliege.be (M.-P.H.); 2Department of Infectious Diseases, Liège University Hospital, 4000 Liège, Belgium; Samira.Azarzar@uliege.be (S.A.); Alyssia.Mesdagh@uliege.be (A.M.); Patricia.Dellot@chuliege.be (P.D.); yasminebelhadj01@gmail.com (Y.B.); Raphael.Schils@chuliege.be (R.S.); Michel.Moutschen@uliege.be (M.M.); 3Laboratory of Cellular and Molecular Immunology, GIGA Institute, University of Liège, 4000 Liège, Belgium; Amandine.Tytgat@uliege.be (A.T.); Fabrice.Bureau@uliege.be (F.B.); 4Immunology-Vaccinology Laboratory, Department of Infectious and Parasitic Diseases, FARAH, University of Liège, 4000 Liège, Belgium; L.Gillet@uliege.be; 5Unit of Animal Genomics, GIGA Institute, University of Liège, 4000 Liège, Belgium; srahmouni@uliege.be; 6Biostatistics and Medico-Economic Information Department, University Hospital of Liege, 4000 Liege, Belgium; M.Ernst@chuliege.be

**Keywords:** COVID-19, SARS-CoV-2, diagnosis, persistence

## Abstract

The testing and isolation of patients with coronavirus disease 2019 (COVID-19) are indispensable tools to control the ongoing COVID-19 pandemic. PCR tests are considered the “gold standard” of COVID-19 testing and mostly involve testing nasopharyngeal swab specimens. Our study aimed to compare the sensitivity of tests for various sample specimens. Seventy-five participants with confirmed COVID-19 were included in the study. Nasopharyngeal swabs, oropharyngeal swabs, Oracol-collected saliva, throat washes and rectal specimens were collected along with pooled swabs. Participants were asked to complete a questionnaire to correlate specific clinical symptoms and the symptom duration with the sensitivity of detecting COVID-19 in various sample specimens. Sampling was repeated after 7 to 10 days (T2), then after 14 to 20 days (T3) to perform a longitudinal analysis of sample specimen sensitivity. At the first time point, the highest percentages of SARS-CoV-2-positive samples were observed for nasopharyngeal samples (84.3%), while 74%, 68.2%, 58.8% and 3.5% of throat washing, Oracol-collected saliva, oropharyngeal and rectal samples tested positive, respectively. The sensitivity of all sampling methods except throat wash samples decreased rapidly at later time points compared to the first collection. The throat washing method exhibited better performance than the gold standard nasopharyngeal swab at the second and third time points after the first positive test date. Nasopharyngeal swabs were the most sensitive specimens for early detection after symptom onset. Throat washing is a sensitive alternative method. It was found that SARS-CoV-2 persists longer in the throat and saliva than in the nasopharynx.

## 1. Introduction

Coronavirus disease 2019 (COVID-19) is an acute infection of the respiratory tract that appeared in late 2019 [1,2]. For some individuals, COVID-19 can cause symptoms that last for weeks or months after the infection has gone [3]. The impact of the COVID-19 pandemic on the health care system has been dramatic [4,5,6]. Sensitive, inexpensive and easy-to-perform diagnostic tests are indispensable for controlling the ongoing COVID-19 pandemic. The global COVID-19 pandemic has seen a sharp rise in interest in the processes and techniques used in laboratories, specifically seeking assurances around access to timely, reliable diagnostic results with high sensitivity (the ability of a test to correctly identify patients with a disease) and specificity (the ability of a test to correctly identify people without the disease). To date, no perfect “gold standard” test is available for the diagnosis of COVID-19. Viral RNA detection by reverse transcription-polymerase chain reaction (RT-PCR) testing continues to play a central role in diagnosis, while rapid antigen detection is limited by its sensitivity [7,8]. The current standard involves testing nasopharyngeal (NP) or oropharyngeal (OR) swab specimens, although saliva (S) or throat wash (TW) may be alternative diagnostic samples [9,10,11]. The latter does not require direct interaction between health care workers and patients, which is considered one of the major sources of testing bottlenecks and nosocomial infections. These samples may therefore be very useful to detect SARS-CoV-2 infection in specific settings, for instance, to test asymptomatic people in collective settings (nursing homes and precarious people in housing structures) or to detect early outbreaks in places where clusters are more likely to occur. Indeed, if test sensitivity is clearly critical, other features of the various testing options should be considered. In particular, the context of how the test is being used is crucial, especially when broad screening is desperately needed. The key question is not only the extent to which RNA or proteins are detected in a single sample, but also how effectively infections are detected in a population through the repeated use of a particular test [12]. Non-traumatic self-collected samples such as saliva or throat washes may be extremely valuable options for massive testing if their sensitivity is shown to be sufficient [13,14].

Here, we propose a rigorous and in-depth comparison of SARS-CoV-2 detection sensitivity between NP swabs, OP swabs, rectal swabs, pooled swabs, S and TW. Longitudinal sampling was performed to assess assay sensitivity over time. As different clinical presentations may be associated with privileged viral replication sites, we correlated the assay sensitivity with symptoms at the time of sample collection to determine whether symptom-based individualized diagnostic strategies would increase sensitivity.

## 2. Method

### 2.1. Participant Enrollment

Patients admitted to Liège University Hospital, Belgium, who tested positive for SARS-CoV-2 (test performed on any clinical specimen, mostly NP) were invited to participate in the study. Additionally, asymptomatic hospital staff members (including health care workers and administrative staff) enrolled in a prospective study implemented as an active monitoring protocol for COVID-19, aiming to detect SARS-CoV-2 infection at the earliest stage, were invited to participate in the study when they tested positive for SARS-CoV-2.

Between 8 October 2020 and 13 December 2020, 61 inpatients with COVID-19 at Liège University Hospital were enrolled in our study. In parallel, 14 staff members agreed to be included in the present study following a COVID-19 diagnosis. Demographics, clinical data and samples were collected after the study participants acknowledged that they had understood the study protocol and signed the informed consent form. The protocol was approved by the ethics committee of Liège University Hospital (approval number 2020-139).

### 2.2. Sample Collection and SARS-CoV-2 Detection

For inpatients, samples were obtained during hospitalization. For outpatients (either staff members or follow-up visits after discharge), appointments for sampling were proposed at the hospital’s COVID-19 center. For self-sampling (S and TW), patients were asked to collect specimens in the morning after waking up but before eating, drinking and tooth brushing. The TW was performed with 5 mL of saline and collected into a sterile collection tube after approximately 30 s of gargling. Saliva (S) was collected using Oracol S14 (Malvern Medical Developments, Worcester, UK), a simple device for the collection of oral fluid designed to be used in a similar manner to a toothbrush [15]. After collection, we added 1mL of Copan UTM-RT transport medium liquid (Mast Group, Brescia, Italia) into the device and centrifuged at 3000 rpm for 10 min. Then, we recovered the medium to proceed with the analysis. The NP and OP swabs were sampled as described previously [16,17]. Rectal (R) sampling was performed using a rectal swab [17]. A pooled swab was acquired, corresponding to NP + OP + R or NP + OP (without rectal sample) when the participant refused the rectal sampling. Participants were also asked to complete a questionnaire to correlate specific phenotypes with the sensitivity of the collection method. Briefly, the questionnaire aimed to determine the timing of symptom onset and to define whether the participant presented symptoms associated with COVID-19 at the time of sample collection [18]. When possible, samples were collected longitudinally, 7 to 10 days later (T2) and 14 to 20 days later (T3). Individuals who agreed to provide samples at the second and third time points of the study were asked to complete the questionnaire again.

Samples were tested for SARS-CoV-2 infection using the Roche Diagnostics Cobas 6800 SARS-CoV-2 test. The whole process was fully automated, from sample preparation to the detection of the amplified genes (ORF1a/b and E).

### 2.3. Statistical Analysis

Data are summarized as the means and standard deviations (SDs), medians and interquartile ranges (IQRs), and extreme values for continuous variables, while frequency tables were used for the categorical variables. Cohen’s kappa coefficient (κ) was calculated to measure the inter-rater reliability of categorical items. A logistic regression analysis was conducted to explore the effects of symptoms on detection sensitivity at T1, and a linear mixed model was performed to study the evolution of viral loads over time based on repeated measures in individual patients. The results were considered significant at the 5% critical level (*p* < 0.05). All calculations and graphs were performed with SAS version 9.4 and R version 3.6.1.

## 3. Results

### 3.1. Sensitivity of Sample Specimens at Diagnosis

Seventy-five subjects with confirmed SARS-CoV-2 infection were included in the study. The median age at sample collection was 66 years. Forty-four percent of participants were female. For each type of sample, the cycle threshold (CT) mean and median values are indicated for positive samples in Table 1. Sample types are indicated in Table 1 based on SARS-CoV-2 detection sensitivity at T1, T2 and T3. Figure 1 presents the sensitivity of SARS-CoV-2 detection in clinical samples at each time point.

At T1, NP swab specimens (84.3%) exhibited the highest sensitivity, similar to pooled swabs (83%), followed by TW (74%), S (68.2%), OP (58.8%) and R (3.5%) (Table 1 and Figure 1). Viral loads increased (cycle threshold (CT) values decreased) together with the assay sensitivity. The median duration of symptoms before sample collection at T1 was 12 days (Table 1).

### 3.2. Sensitivity of Sampling Methods over Time

The levels of SARS-CoV-2 RNA decreased after symptoms onset and differed according to sample specimens. Individual evolutions of loads (CT values) over time are represented for each sampling method in Figure 2. To consider the correlation between samples collected from the same person, a linear mixed-effects regression model with interaction was performed (Table 2). The model induced a different time effect between sample specimens. The evolution over time for each sampling method was estimated (Appendix A) and is illustrated in Figure 2. A pairwise comparison (Scheffé correction) concluded that NP, Pool or TW samplings were associated with significantly larger CT values than OP or S sampling.

We observed that the viral loads measured on NP decreased faster than those in other sample specimens (Appendix A, Figure 2). Accordingly, the sensitivity of NP drastically decreased to reach 27.5% and 10% at T2 and T3, respectively (Table 1, Figure 1). By contrast, the decrease in the levels of SARS-CoV-2 RNA was slower for other specimens, in particular for TW, OP and S (Appendix A, Figure 2). Consequently, TW had the highest sensitivity at T2 and T3, with 47.6% and 50% of positive samples, respectively (Figure 1 and Table 1).

### 3.3. Interspecimen Reliability Measurements

Interspecimen reliability measurements at the first time point are listed in Table 2. Cohen’s kappa was calculated to measure the agreement between two sample types. Interestingly, the level of agreement was low between NP swab specimens and TW (Table 2 and Appendix A). Most of the patients with a negative result from NP swabs were positive when TW was analyzed (7 out of 11, Appendix A). The level of agreement was much higher between TW, OP and S (Table 2).

### 3.4. Correlation between Symptoms and SARS-CoV-2 Detection

Assay sensitivity was associated with symptomatology. We determined the percentage of positivity depending on the presence of specific symptoms associated with COVID-19 [18] (Table 3). The percentages of SARS-CoV-2 detection through NP swabs were higher when patients experienced dyspnea, diarrhea, abdominal pain, headaches or myalgia. The percentages of SARS-CoV-2 detection through TW were higher when patients experienced anosmia, ageusia, rhinorrhea, fever or sore throat (Table 3). In particular, the probability of detecting SARS-CoV-2 was significantly higher (90.5%) in participants with rhinorrhea (OR 4.89, *p* = 0.047) (Appendix A). SARS-CoV-2 infection was best detected in patients with fever as one of the symptoms through NP, TW or S sampling. Indeed, the probability of detecting SARS-CoV-2 in Oracol-collected saliva was higher (93.3%) in this group of patients (OR 9.03, *p* = 0.041) than in the other groups (Appendix A). Infections in patients with sore throat were often detected with TW (93.3%) (OR 6.63, *p* = 0.078). In contrast, the probability of detection with the pooled sample decreased in participants with sore throat (Appendix A).

## 4. Discussion

Since the identification of COVID-19 in China, millions of people have been infected in numerous countries, defining the disease as a pandemic. Public health responses have varied substantially between regions. Most countries implemented interventions based on large-scale testing and measures, including wearing masks, quarantine and social distancing. To date, testing for SARS-CoV-2 infection has mostly relied on NP swab sampling. Although alternative approaches have been proposed, some of them are easy to perform without the assistance of health care workers, including saliva or throat wash samples. The self-collection of saliva or throat wash also alleviates the demands for supplies of swabs and personal protective equipment [9,14].

NP swab specimens have been shown to be superior to OP swab specimens for the detection of SARS-CoV-2 [16]. Indeed, Xiong Wang and colleagues reported a higher sensitivity of NP, with as many as 73.1% of NP-positive cases being negative in oropharyngeal swabs [16]. In contrast, throat washes have produced better results, although the number of participants was very low in these studies [10,19]. Indeed, Wen-Liang Guo and colleagues reported that the positive test rate of TW was much higher than that of NP swabs. However, only eleven patients were included in the study, and no more than twenty-four paired throat washes and NP swabs were analyzed [10]. Samples were also collected very late during the course of the disease (at a median of 53 days after symptom onset), making the results difficult to interpret. Saliva specimens were recently shown to provide comparable results to NP swab specimens, although the sensitivity of saliva specimens compared to NP swabs varied among studies [9,11,20]. Differences were likely linked to various methods of saliva collection (using a specific device or not), cohort specificity (in- or outpatients) and sample processing. Pre-analytic differences are probably crucial to interpret saliva specimen sensitivity. In addition, it should be mentioned that gargling and saliva collection require the active cooperation of patients. This could be an issue, particularly in hospitalized patients.

To our knowledge, our study is the first to propose an in-depth longitudinal comparison of various specimen options available to detect SARS-CoV-2 in the same individuals. Overall, we showed that NP swabs were the most sensitive specimens when performed early after symptom onset, although throat washes showed good sensitivity as well. We further revealed a poor correlation between NP and TW, suggesting that a patient suspected of having COVID-19 who tested negative for SARS-CoV-2 through NP may indeed be false negative and should be tested using throat washing.

We documented that the duration of symptoms mainly affected the sensitivity of detection using NP swabs. The sensitivity of SARS-CoV-2 detection performed using NP swabs drastically decreased with symptom duration. At the second and third time points of our study, throat washing was the most sensitive sampling method. We determined that throat washing was more sensitive than NP at more than 21 days after symptom onset, which should be considered to improve SARS-CoV-2 testing strategies.

Finally, the sensitivity of the sample specimen varied depending on the clinical presentation. SARS-CoV-2 detection in Oracol-collected saliva specimens was lower than that in NP or TW samples but was higher in patients with fever at sample collection. SARS-CoV-2 detection in throat wash specimens was particularly high in patients with rhinorrhea, a phenomenon that might be due to overnight discharge through the throat. Larger studies are needed to confirm that symptom-based testing strategies are also an interesting option.

Differences between our study and previously reported results are explained by several factors. First, saliva samples were collected with a specific device in our study. Second, we mostly included hospitalized participants, some of whom had very low saliva production, particularly older mouth-breathing individuals. Third, we showed that the saliva specimens were affected by symptom duration, which was long in our study. We indeed observed that patients often came to the hospital late in the course of the infection.

Our study has several limitations. Our results should be interpreted with caution as sensitivity is affected by both the analytical method and the pre-analytical steps. The use of other methods might yield a different outcome. Moreover, the sensitivity of the method was only evaluated and validated using SARS-CoV-2 viral culture spiked in negative nasopharyngeal swab samples.

In conclusion, our study showed that NP swab specimens should remain the preferred specimen to detect SARS-CoV-2 infection early after symptom onset. However, alternatives can be used with good efficacy—particularly throat washes—in suspected patients with negative detection using NP swabs, or in settings where the involvement of health care workers is a major bottleneck to efficient testing. Throat wash specimens also appear to be the best option for individuals presenting late during the course of the disease. The detection of SARS-CoV-2 in Oracol-collected saliva specimens was disappointing in our study, with the noteworthy exception of participants presenting with fever. Finally, the finding of higher SARS-CoV-2 detection in throat samples at later time points also has implications for viral persistence studies, with a potential critical effect on quarantine duration and PCR-based lifting of the quarantine strategies. Culture experiments are needed to confirm the infectivity of the virus-identified throat wash samples.

## Figures and Tables

**Figure 1 pathogens-10-01362-f001:**
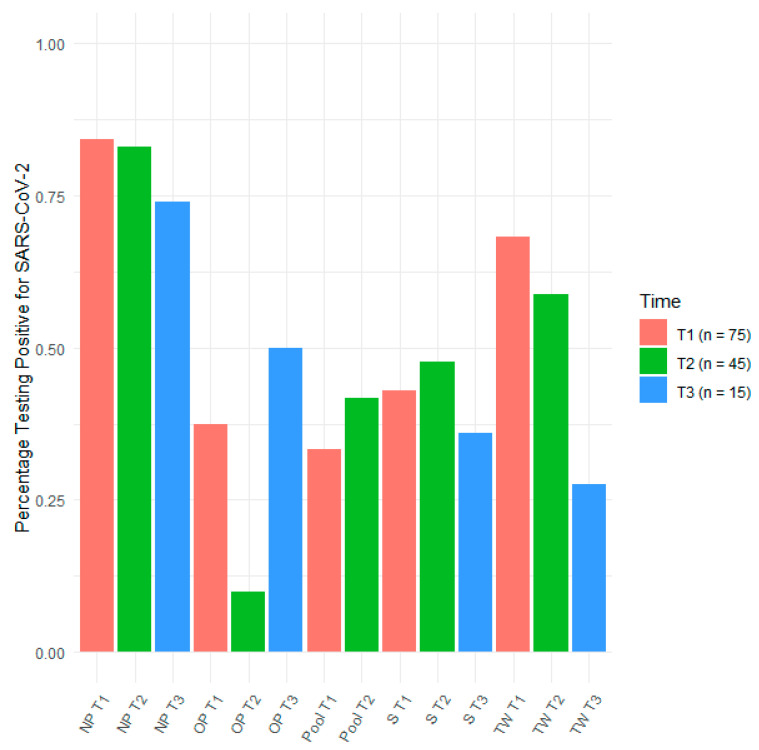
Sensitivity of sample specimens. Percentage of positive tests per sample type at T1 (n = 75), T2 (n = 45) and T3 (n = 15). The percentage of pooled samples at T3 is not indicated, as only one sample was available.

**Figure 2 pathogens-10-01362-f002:**
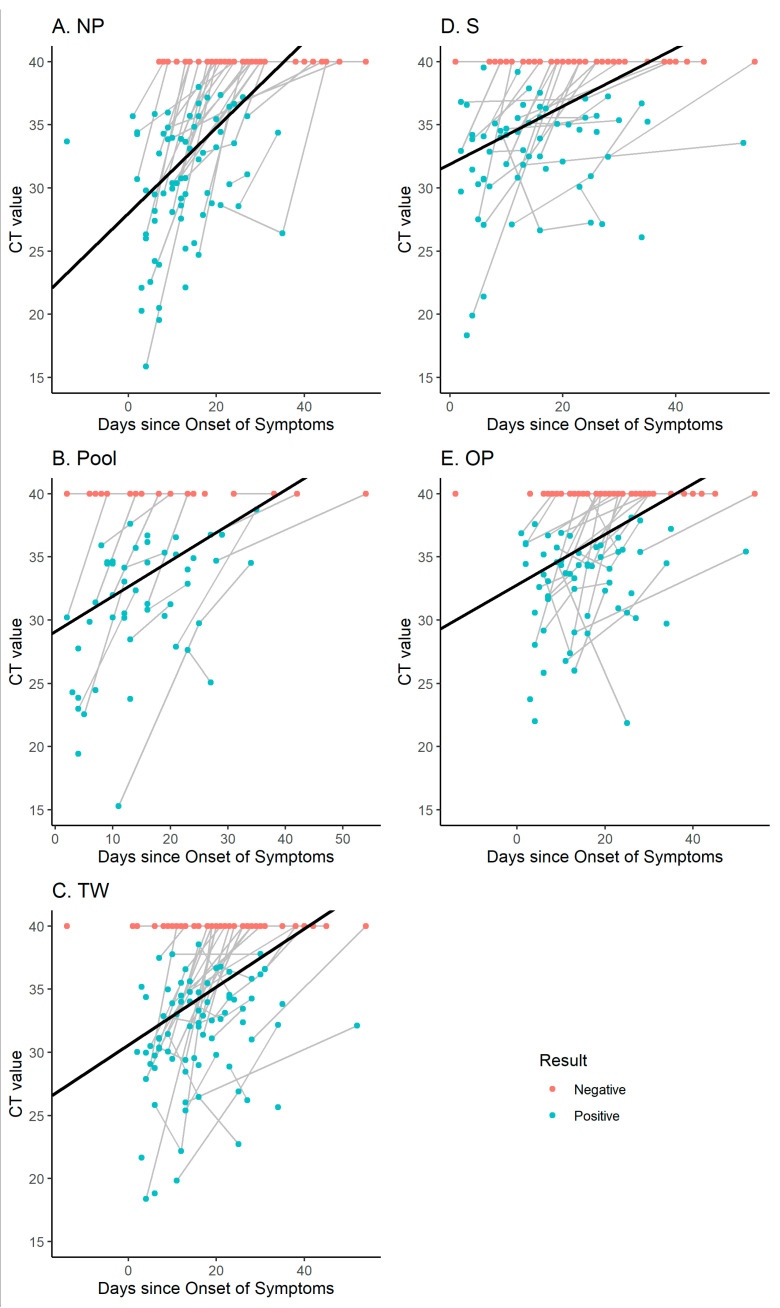
Evolution of CT values over time. Evolution of viral load (CT values) over time. Negative samples are represented with red dots (CT value of 40) and positive samples are represented with blue dots. Samples from the same participants are connected with gray lines. The estimated mixed model is represented with black curves according to sampling methods: nasopharynx swabs (NP, **A**), Pool (**B**), throat wash samples (TW, **C**), Oracol-collected saliva samples (S, **D**) and oropharynx swabs (OP, **E**).

**Table 1 pathogens-10-01362-t001:** Patient characteristics and sensitivity of sample types.

Variable	N	N (%)	Mean ± SD	Median (Q1–Q3)	Extremes
Age (years)	75		63.2 ± 16.7	66 (53; 77)	24; 96
Sex, Female	75	33 (44.0)			
Positive swab at T1					
NP	70	59 (84.3)	29.7 ± 5.5	30.3 (26.3; 34.3)	13.7; 38.0
Pool	47	39 (83.0)	30.7 ± 5.4	31.3 (27.8; 34.7)	15.3; 37.6
TW	73	54 (74.0)	30.8 ± 4.4	31.1 (29.1; 34.0)	18.4; 37.8
S	66	45 (68.2)	32.4 ± 4.6	33.9 (30.7; 35.1)	18.3; 39.5
OP	68	40 (58.8)	32.7 ± 3.9	34.3 (30.4; 35.7)	22.0; 37.6
R	29	1 (3.5)	34.8		
TW	42	20 (47.6)	32.7 ± 4.5	33.9 (29.6; 36.0)	22.2; 38.6
Pool	21	9 (42.9)	32.6 ±3.8	32.9 (31.2; 34.9)	25.1; 38.8
OP	40	15 (37.5)	33.1 ± 2.9	33.7 (30.6; 35.4)	27.4; 38.1
S	39	14 (35.9)	33.1 ± 3.4	33.3 (30.9; 35.6)	26.6; 37.9
NP	40	11 (27.5)	33.0 ± 3.7	33.5 (29.2; 36.4)	26.4; 37.4
Positive swab at T3					
TW	14	7 (50.0)	32.6 ± 4.6	34.2 (32.2; 35.8)	22.7; 36.2
OP	12	5 (41.7)	32.6 ± 6.2	34.5 (33.0; 35.6)	21.9; 37.9
S	12	4 (33.3)	34.2 ± 4.7	36.1 (31.4; 37.0)	27.2; 37.3
NP	10	1 (10.0)	34.4		
Pool	1	1 (100.0)	34.5		
Days between symptom onset and sample collection at T1	75		12.9 ± 8.8	12 (6; 18)	−14; 35
Days between symptom onset and sample collection at T2	45		24.5 ± 11.4	23 (16; 28)	8; 54
Days between symptom onset and sample collection at T3	15		28.0 ± 7.2	28 (22; 31)	19; 45
Number of symptoms at T1	73		4.1 ± 2.1	4 (3; 6)	0; 9
Number of symptoms at T2	28		3.1 ± 1.9	3 (1.5; 4.5)	0; 7
Number of symptoms at T3	11		2.1 ± 2.0	2 (0; 4)	0; 5

SD: Standard deviation; Q1–Q3: First and third quartiles; T1: First time-point; NP: Nasopharynx sample; TW: Throat wash; S: Saliva; OP: Oropharynx sample; R: Rectal sample.

**Table 2 pathogens-10-01362-t002:** Inter-rater reliability measurements at T1.

	NP	Pool	TW	S	OP	R
**NP**		κ = 0.25 [−0.10; 0.59] *p* = 0.092	κ = 0.078 [−0.16; 0.32] *p* = 0.50	κ = 0.29 [0.06; 0.53] *p* = 0.0096	κ = 0.15 [−0.05; 0.35] *p* = 0.14	κ = 0.012 [−0.01; 0.04] *p* = 0.68
**Pool**			κ = 0.044 [−0.23; 0.32] *p* = 0.75	κ = 0.32 [0.01; 0.63] *p* = 0.028	κ = 0.050 [−0.19; 0.29] *p* = 0.68	κ = 0.022 [−0.02; 0.07] *p* = 0.59
**TW**				κ = 0.48 [0.25; 0.72] *p* < 0.0001	κ = 0.46 [0.25; 0.67] *p* < 0.0001	κ = 0.047 [−0.05; 0.14] *p* = 0.41
**S**					κ = 0.50 [0.28; 0.72] *p* < 0.0001	κ = 0.042 [−0.04; 0.12] *p* = 0.46
**OP**						κ = −0.071 [−0.21; 0.07] *p* = 0.31

NP: nasopharynx sample; TW: Throat wash; S: Saliva; OP: Oropharynx sample; R: rectal sample

**Table 3 pathogens-10-01362-t003:** Participant symptoms.

Symptoms		T1 (n = 75)		T2 (n = 45)		T3 (n = 15)	
		N	N (%)	N	N (%)	N	N (%)
Cough		73	47 (64.4)	28	9 (32.1)	10	3 (30.0)
Dyspnea		73	39 (53.4)	28	13 (46.6)	10	4 (40.0)
Sore throat		73	15 (20.6)	28	3 (10.7)	10	0 (0.0)
Rhinorrhea		73	22 (30.1)	28	8 (28.6)	10	2 (20.0)
Stomach aches		73	15 (20.6)	28	0 (0.0)	10	0 (0.0)
Fever		73	16 (21.9)	28	1 (3.6)	11	0 (0.0)
Diarrhea		73	17 (23.3)	28	1 (3.6)	10	1 (10.0)
Muscle pain		72	30 (41.7)	28	8 (28.6)	10	0 (0.0)
Ageusia		73	17 (23.3)	28	10 (35.7)	10	3 (30.0)
Anosmia		73	19 (26.0)	28	9 (32.1)	10	3 (30.0)
Headache		73	28 (38.4)	28	11 (39.3)	10	1 (10.0)
Other symptoms		73	37 (50.7)	28	13 (46.4)	10	6 (60.0)

## Data Availability

Data is contained within the article or Appendix A.

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
