# Peer review of "In-Depth Longitudinal Comparison of Clinical Specimens to Detect SARS-CoV-2"

_pathogens, 2021, doi:10.3390/pathogens10111362_

Round 1

Reviewer 1 Report

Authors, Defeche et al., present here an in-depth longitudinal comparison of various clinical specimens to detect SARS-CoV-2 infection. Given the need for improved and continued monitoring of the spread of various strains of SARS-Cov-2, this study presents very clear and concise information on the use of various clinical specimens over time, with a need to minimize invasive and 'healthcare worker needed' sample collection. Schools across globe schedule to commence reopening and in-person classes, this study process equally sensitive SARS-CoV-2 testing that can use TW or S as the biological sample with a room for pooling, without loss in sensitivity. 

MINOR COMMENTS

1) I would rephrase the very first sentence as following, 'Testing and isolation of COVID-19 patients are necessary to control the ongoing COVID-19 pandemic.'

2) Delete, 'Indeed' from the last sentence in the results section of the abstract. 

3) Improved resolution for Figure 2. Found it VERY hard to read and follow the image. Greatly appreciate it if authors could keep in mind moving forward for all publications that increased font and clear images are helpful while reviewing articles. 

Author Response

REVIEWER 1.

Authors, Defeche et al., present here an in-depth longitudinal comparison of various clinical specimens to detect SARS-CoV-2 infection. Given the need for improved and continued monitoring of the spread of various strains of SARS-Cov-2, this study presents very clear and concise information on the use of various clinical specimens over time, with a need to minimize invasive and 'healthcare worker needed' sample collection. Schools across globe schedule to commence reopening and in-person classes, this study process equally sensitive SARS-CoV-2 testing that can use TW or S as the biological sample with a room for pooling, without loss in sensitivity. 

We thank the reviewer for the positive comments.

MINOR COMMENTS

- I would rephrase the very first sentence as following, 'Testing and isolation of COVID-19 patients are necessary to control the ongoing COVID-19 pandemic.'

It has been changed. The entire manuscript has been edited by the SNAS Team, as requested by other reviewers.

- Delete, 'Indeed' from the last sentence in the results section of the abstract. 

This has been done.

- Improved resolution for Figure 2. Found it VERY hard to read and follow the image. Greatly appreciate it if authors could keep in mind moving forward for all publications that increased font and clear images are helpful while reviewing articles. 

We agree with the reviewer. We have made considerable efforts to change the way of presenting the data, as requested. The quality of files has been increased as well.

Reviewer 2 Report

There is a very interesting idea at the core of this manuscript, and I would encourage the authors to take the time to extensively redraft this manuscript. At present it is confusing and poorly written.

As examples:

Line 28-29 - whilst I understand what the authors are gesturing towards, it is a confusing sentence that needs rewriting.

Line 28-29 - This is poor English.

Line 40 - The authors have not defined RT-qPCR adequately.

Line 40 - The authors have not defined which rapid antigen test they are referring to.

Line 43 - 45 - This is poor English.

Line 47 - I have no idea what Collectivities are. The authors have not defined this term.

Line 47-48 - 'Precarious population in housing structures' This is a meaningless phrase.

These are simply examples taken from the first half of the manuscript - due to the sheer amount of the text that is unreadable I have not documented the rest. I would strongly urge the authors to get a proof-reader to advise on basic English.

In addition; the introduction does not adequately explain the importance of  sensitivity, selectivity etc to the readers. This is a problem as the manuscript is predominantly dealing with this issue - some groundwork is needed. 

The methods are adequate, but again, would benefit from a rewrite for clarity, especially regarding the participant enrolment (eg; Line 77 - it is assumed 'out-patients' refers to follow-up visits, or maybe the staff members sampled?)

The authors should be explicit around why they chose to use CT value rather than viral load.

Figure 1 is difficult to parse, but intelligible. Figures 2 and 3 are completely unintelligible and I would urge the authors to rethink the way they have chosen to present the data.

Author Response

REVIEWER 2. 

There is a very interesting idea at the core of this manuscript, and I would encourage the authors to take the time to extensively redraft this manuscript. At present it is confusing and poorly written.

As examples:

Line 28-29 - whilst I understand what the authors are gesturing towards, it is a confusing sentence that needs rewriting.

Line 28-29 - This is poor English.

Line 40 - The authors have not defined RT-qPCR adequately.

Line 40 - The authors have not defined which rapid antigen test they are referring to.

Line 43 - 45 - This is poor English.

Line 47 - I have no idea what Collectivities are. The authors have not defined this term.

Line 47-48 - 'Precarious population in housing structures' This is a meaningless phrase.

These are simply examples taken from the first half of the manuscript - due to the sheer amount of the text that is unreadable I have not documented the rest. I would strongly urge the authors to get a proof-reader to advise on basic English.

This has been corrected as requested. Our manuscript has been edited by The SNAS Team (Springer Nature Author Services) in order to improve the quality of the writing.  This certificate was issued on September 23, 2021 and may be verified on the SNAS website using the verification code 656A-E2B8-B51E-C33E-3074.

In addition; the introduction does not adequately explain the importance of sensitivity, selectivity etc to the readers. This is a problem as the manuscript is predominantly dealing with this issue - some groundwork is needed. 

We modified the introduction as requested.

The methods are adequate, but again, would benefit from a rewrite for clarity, especially regarding the participant enrolment (eg; Line 77 - it is assumed 'out-patients' refers to follow-up visits, or maybe the staff members sampled?)

We have modified the method section to make it clearer.

The authors should be explicit around why they chose to use CT value rather than viral load.

CT value has been commonly used in the recent literature on COVID-19 (PMID: 34264527; PMID: 34601145; PMID: 34592917). It is inversely correlated to viral load.

The Roche Diagnostics Cobas 6800 SARS‐CoV‐2 is a qualitative assay (positive-negative results) giving access to the Ct values for the targets genes. Though the sensitivity of the method has been assessed during the validation process (see reviewer 3), results are given in a qualitative way and Ct values are used as no calibrator is provided by the manufacturer.

Figure 1 is difficult to parse, but intelligible. Figures 2 and 3 are completely unintelligible and I would urge the authors to rethink the way they have chosen to present the data.

We agree with the reviewer. We have made considerable efforts to change the way of presenting the data, as requested. The quality of files has been increased as well.

Reviewer 3 Report

In the abstract rectal specimens are mentiones, but no results from them.
saliva is a wide term. It should be explained already in the abstract, how these samples were collected.
very important - this was not passive drooling or spitting. the website the authors have cited does not work.
nevertheless, the oracol device is not much different from salivette from Sarstedt. in our hands, saliva does contain the virus, saliva collected using salivette does not.
it might be that the cells with the virions are mostly in the sponge and not in the analyzed liquid.
this is of huge importance regarding the interpretation of the results and should be reflected in the text - in the intro, in the methods, but especially in the discussion and in the abstract.
the time points seem not to be defined, but should at least approximately described already in the abstract.
technical sensitivity should be reported with spiked samples.
how was covid-19 positivity defined, what were the inclusion criteria, if 15% of patients were not positive in the nasopharynx?
"Throat washing is an easy-to-perform, cheap and sensitive alternative method" is a true statement, but it is not a conclusion from this study.
"A pooled swab was acquired, corresponding to NP+OP+R or NP+OP when the participant refused rectal sampling" - this is very unclear to me...
the sensitivity is affected by the chosen analytical method and the pre-analytical steps. this means that the results are limited to the Roche system.
using other methods for RNA isolation and subsequent analysis might yield a different outcome. this should be mentioned...
also, there are methods described in the literature for molecular detection without RNA isolation - LAMP on saliva samples for example.
gargling and saliva collection requires active cooperation of patients, which is not easy sometimes. this should also be mentioned and discussed.
the time dependence should be shown on those patients/probands who had all three samples. otherwise, there is bias...
the regression must be shown with individual values
the graphs should use colors with caution...

Author Response

REVIEWER 3.

In the abstract rectal specimens are mentiones, but no results from them.

We have added the result in the abstract.

saliva is a wide term. It should be explained already in the abstract, how these samples were collected.

It has been added in the abstract.

very important - this was not passive drooling or spitting. the website the authors have cited does not work.
nevertheless, the oracol device is not much different from salivette from Sarstedt. in our hands, saliva does contain the virus, saliva collected using salivette does not.
it might be that the cells with the virions are mostly in the sponge and not in the analyzed liquid.
this is of huge importance regarding the interpretation of the results and should be reflected in the text - in the intro, in the methods, but especially in the discussion and in the abstract.

This is an important point raised by the reviewer. We agree that we should be more specific, in particular in the method section, but also in the discussion.

We modified our manuscript in order to make it clearer. In particular, we have included additional information regarding the pre-analytic process.

Saliva was collected using Oracol (Malvern Medical Developments, Great Britain) following the manufacturer's instructions. After collection, we added 1mL of Copan UTM-RT transport medium liquid (Mast Group, Great Britain) into the device and centrifuged at 3000rpm for 10 minutes. Then, we recovered the medium to proceed with the analysis.

The discussion has also been expanded. The abstract has been modified as well.

We do not understand why the reviewer did not access the website as the link is valid.

the time points seem not to be defined, but should at least approximately described already in the abstract.

This has been included in the abstract.

technical sensitivity should be reported with spiked samples.

This is a good point raised by the reviewer. The sensitivity of the method has been evaluated using SARS-CoV-2 viral culture (provided by the NRC-KUL, Belgium) spiked in negative nasopharyngeal swab samples. The limit of sensitivity (100% detection rate) has been estimated to be 1 cp/mL. We did not evaluate the sensitivity of the method on other sample types (saliva, gargling, ...). We acknowledge that it is a limit of our study. It has been included in the discussion (section limitations).

how was covid-19 positivity defined, what were the inclusion criteria, if 15% of patients were not positive in the nasopharynx?

COVID-19 positivity was defined as having a positive SARS-CoV-2 PCR test. This could be performed on any clinical specimen (mostly NP or TW). Following inclusion, all clinical specimens were collected at the same time to be comparable. It means that a participant diagnosed with a NP test had a second NP test performed at the same time than other specimens. We modified the method section to make it clearer.

"Throat washing is an easy-to-perform, cheap and sensitive alternative method" is a true statement, but it is not a conclusion from this study.

We agree with the reviewer. We modified this sentence in the abstract.

"A pooled swab was acquired, corresponding to NP+OP+R or NP+OP when the participant refused rectal sampling" - this is very unclear to me...

Some participants refused the rectal sampling. In that case, the pooled sample is a combination of NP+OP and does not include a rectal specimen. We modified the method section for clarity.

the sensitivity is affected by the chosen analytical method and the pre-analytical steps. this means that the results are limited to the Roche system.
using other methods for RNA isolation and subsequent analysis might yield a different outcome. this should be mentioned...
also, there are methods described in the literature for molecular detection without RNA isolation - LAMP on saliva samples for example.

We agree with the reviewer. We have added a section regarding the limits of our study in the discussion in order to discuss the important comments made by the reviewer.

gargling and saliva collection requires active cooperation of patients, which is not easy sometimes. this should also be mentioned and discussed.

This has been added in the discussion.

the time dependence should be shown on those patients/probands who had all three samples. otherwise, there is bias...

We do not agree with the reviewer. A linear mixed-effect regression model was performed to analyze the time and sampling dependences by considering repeated measures (different samples and/or time points) for each patient. This method allows unbalanced design without introducing any bias.

In addition, performing the analysis only on those who had all three samples would induce a bias as it would select patients hospitalized for longer time and/or staff members who had an easy access to the hospital.

the regression must be shown with individual values

We have modified the figures as requested.

the graphs should use colors with caution...

The graphs have been modified.

Round 2

Reviewer 2 Report

The authors have addressed all concerns raised, and have raised the quality of the manuscript.

Reviewer 3 Report

the authors have followed some (but not all) of my suggestions and improved the manuscript. so, I pick some of the leftovers that have to be included.

fist time point, later time points... this tells the reader nothing.
please, specify. if the time points are 7-10 days apart, then the third time point is 14-20 days later why cannot this be clearly stated? especially in the abstract

the authors have to state that it is not whole saliva that they have collected, rather they should mention the name of the collection device. in all occasions. call it Oracol collected saliva, for example.

this is the link from the references
https://www.malmed-oracol.co.uk/oracol/
does it work? not in my browser...

sensitivity of  1 copy per 1 ml? I am not convinced. what was the Ct value? what is the volume used for isolation? 

if the first nasopharyngeal test was negative and the patient was included... which test was positive? this group of patients was different - what was the difference? next sampling was positive? or were the subsequent sampling negative as well? were these patients in later stages of the disease?

the authors think that they do not introduce bias when analyzing all patients, including those with missing samples. they state that the exclusion of patients might prefer those who were hospitalized. indeed, that is, however, also the explanation for the bias that is introduced when missing samples are ignored.
if the major outcome is the dynamic change of sensitivity, then this should be analyzed on those patients who provided samples at all time points. otherwise there is bias, no matter which statistical test is used. what if all patients who do not have the second or third sample died? or what if they had no symptoms? then the first time point would include them, but the later time points would not and the whole difference in sesitivities would be due to different patient populations. so, the analysis has to be performed on patients attending all time points. maybe in addition to the current analysis. otherwise, the interpretation is misleading.
